# Joint Decision on Pricing and Ordering for Omnichannel BOPS Retailers: Considering Online Returns

**Jinrong Liu**  **and Qi Xu** *

Glorious Sun School of Business and Management, Donghua University, Shanghai 200051, China; liujinrongdhu@163.com

* Correspondence: xuqi@dhu.edu.cn; Tel.: +86-021-6237-8860

**Abstract:** Uncertainty in demand and high online return rates tend to generate a large inventory backlog after a hot selling season. Some well-known companies such as Burberry and H&M choose to burn these backlogs, which is very unfavorable to the sustainable development of enterprises and society. The omnichannel BOPS (Buy Online and Pickup in Store) provides consumers with a new and convenient choice of purchasing channels. Considering the probability of online return, we build expected profit models for retailers before and after opening a BOPS channel based on consumer surplus and purchase ratio via an online channel, store channel, and BOPS channel. Then, we prove the existence of optimal solutions and obtain the joint optimization decision on pricing and ordering. The results show that, under certain conditions, when the proportion of online channel buyers increases, the retailers' optimal decision before opening a BOPS channel is to reduce the price and increase the order quantity, and the retailers' optimal decision after opening a BOPS channel is to simultaneously reduce the price and the order quantity. Whether or not a BOPS channel is opened, when the purchase proportion of store channel increases, the optimal decision of retailers is to increase the price and order quantity at the same time. Furthermore, when the online return rate, the cost of online shopping, the inconvenient cost of the store channel, and the inconvenience of the BOPS channel relative to store channel increase, the optimal decision of retailers is to simultaneously reduce the price and order quantity. Moreover, when the maximum psychological value of the product increases, the optimal decision is to reduce the order quantity while increasing the price. In addition, the opening of a BOPS channel increases optimal price, optimal order quantity, and maximum expected profit.

**Keywords:** BOPS channel; pricing; ordering; consumer surplus; return rate

## 1. Introduction

With the rapid development of mobile Internet and O2O (Online to Offline) e-commerce, many retailers have realized the need to integrate their existing channels to enrich customer value proposition and improve operational efficiency. As a result, there is an emerging focus on "omnichannel retailing" with the goal of providing customers with a seamless shopping experience through all available shopping channels [1–3].

The so-called omnichannel means that, in order to meet the needs of consumers at any time, any place, and any way, the retailer integrates a store channel, an online channel, and a mobile internet channel to sell their products, so as to provide their customers with a seamless buying experience [4]. Among them, the omnichannel retail mode BOPS (Buy Online and Pickup in Store) is becoming an important platform and an organic part of omnichannel retail between merchants and consumers. According to the retailers report of Forrester Research, this BOPS model allows customers to buy online and pick up goods in stores and is considered one of the most important

models in the omnichannel [5]. As of June 2013, 64% of retailers implemented BOPS, such as Wal-Mart, Best Buy, Uniqlo, and Target, according to Retail Systems Research [6]. BOPS can enable customers to experience real-time refined services for online channel shopping, avoid changes in shipping and delivery, and enjoy the convenience of worry-free shopping [7]. In addition, the BOPS model is also a way for retailers to contact new customers, which can generate new physical store transactions and increase sales [8]. According to UPS research, 45% of the customers who choose to pick up from the store will add new orders when they pick up the goods from the store [9]. The 2017 China "Double 11" (November 11) research report issued by the E-commerce Center pointed out that, in 2017, Tmall's online and offline integration trend of "Double 11" accelerated. Suning Online Market, Uniqlo, Inman, and other brand retailers started the two-line promotion mode, which led to a substantial growth of offline stores. For example, Suning Online Market omnichannel achieved a remarkable growth of 163%. Eifini reached 509 offline shipping stores with sales of 152 million, an increase of 52%. The "Double 11" shopping festival has become the largest national shopping festival in China [10].

In order to better meet customers' purchase demands during the hot selling season, brand retailers need to determine the order quantity in advance, especially for large holidays such as "Double 11" in China and "Black Friday" in the United States. Many retailers started to reserve products at least one month or even half a year beforehand. Adequate stocking can avoid sales losses due to shortages, but the high online return rate and uncertain demand often lead to a large inventory backlog after the hot selling holidays, bringing multiple losses to retailers, consumers, and the online shopping market. In China, the National Retail Federation (2014) pointed out that the return rates of online channels are typically between 20% and 40%, with poor fit cited as the main reason. According to David Sobie, co-founder and CEO of Happy Returns [11], about 5–10 percent of in-store purchases are returned, but that rises to 15–40 percent for online purchases. In addition, 72% of enterprises cover the cost of handling returns [12], which makes returns a significant part of the company's cost. The 2018 Consumer Returns in the Retail Industry report stated that total merchandise returns account for nearly $369 billion USD in lost sales for US retailers. This is close to the market cap value of Facebook (November 2018). According to a BBC report, Burberry burned 28.6 million pounds of slow-moving goods in 2017 alone, and the total value of slow-moving products destroyed in the past five years has exceeded 90 million pounds. H&M has also burned 60 tons slow-moving clothing. The behavior of these brand companies in handling inventory not only affects their reputation but also results in a higher loss of profits, and it is not conducive to the sustainable development of the company and the social environment. Obviously, the joint optimization decision of enterprise pricing and ordering has important significance for the sustainable development of enterprises and society.

Therefore, in this paper, we focus on the following three research questions:

1. Before and after opening an omnichannel BOPS channel, do brand retailers have optimal joint decisions on pricing and ordering?
2. What kind of joint pricing and ordering decisions can meet consumer demand while avoiding a large inventory backlog and thereby obtain the maximum benefits?
3. When considering online returns, is it profitable for brand retailers to open a BOPS channel?

To address these questions, we establish expected profit models for brand retailers before and after opening a BOPS channel based on consumer surplus and purchase ratio under different channels. Next, we prove the existence of optimal joint solutions and obtain optimal pricing and ordering joint decisions under certain conditions. Then, we verify them with numerical calculations and obtained useful conclusions.

The main contributions of this paper are as follows. First, considering the online return rate (here, the probability of return is used to describe the return rate), based on the purchase behavior of consumers via an online channel, store channel, and BOPS channel, we design the ratio of consumer purchases in each channel from the perspective of consumer surplus. This is in line with the actual situation of customer consumption behavior and high online return rate. Second, we study brand retailers'

optimal joint decisions on pricing and ordering before and after opening a BOPS channel and make a comparative analysis, which not only satisfies consumers' purchase needs and improves customer loyalty but also minimizes inventory backlog, saves social resources, and reduces waste, thereby maximizing retailers' profits and promoting the sustainability of enterprises and social development. Third, the research theme, perspective, and methods of this study provide a new direction and ideas for sustainable literature research. Therefore, this study has important theoretical significance for the joint optimization decision of pricing and ordering of brand retailers' BOPS omnichannel and provides a reference for omnichannel operation practices and the sustainable development of enterprises.

This remainder of this paper is structured as follows. Section 2 reviews the relevant literature on operation management for an omnichannel BOPS and joint decision on pricing and ordering. Section 3 builds the expected profit models before and after opening a BOPS channel when considering online returns. Section 4 analyzes the influence of different factors on the impact of the optimal decisions and maximum profit. Section 5 offers conclusions and management implications.

## 2. Literature Review

This section sheds light upon two streams of literature. The first stream addresses the omnichannel BOPS, and the second stream addresses the joint decisions on pricing and ordering.

### 2.1. Literature on Omnichannel BOPS

At present, omnichannel BOPS operation management has attracted a lot of attention from the academic community. For example, Gallino and Moreno [13] empirically studied the impact of the BOPS model on retailers' sales via online and offline channels. Kong et al. [14] and Cao et al. [15] analyzed the impact of this model on the price, demand, and profit of retailers when the online price and the store price are the same or different. Gao and Su [7] analyzed the impact of the BOPS model on physical store inventory decisions. Fan et al. [16], Yan and Li [17], and Liu et al. [18] discussed BOPS pricing and service decisions of the supply chain or the retailer. Furthermore, Liu and Zhou [19] discussed the influence of the traditional consumers' proportion and consumers' service sensitivity on the implementation of a BOPS. From the perspective of the differentiation of offline retailer service costs, Liu et al. [20] studied two types of BOPS online and offline channel integration issues in which unit compensation and sales are included offline. Yan et al. [21] found that products with high return rates are not suitable for traditional physical retailers to implement BOPS but may benefit dual-channel retailers in implementing BOPS. Jin et al. [22] found that the ratio of unit inventory cost to the arrival rate of BOPS customers is the key factor to determine the size of BOPS service area. MacCarthy et al. [23] proposed a strategy for physical stores to ensure customer service levels for online orders. In addition, Kim et al. [24] indicated that the consumer perceptions of relative advantage, complexity, compatibility, and risks involved in online shopping are important antecedents for the intention to use BOPS. Paul et al. [25] studied the benefit of ex-plotting any spare capacity in the vehicles replenishing store inventories to reduce online order fulfillment cost by transferring online orders to these vehicles at one or more of the stores visited. Glaeser et al. [26] empirically studied the spatiotemporal location problem motivated by an online retailer that uses the BOPS fulfillment method.

Obviously, these studies on BOPS are mainly about the impact of opening a BOPS on enterprises, conditions of implementation, pricing and service decisions, etc. They did not consider the actual problem of improper ordering before the hot sale period, which may cause a large amount of inventory backlog and make corresponding pricing and ordering decision.

### 2.2. Literature on Pricing and Ordering Decision under Non-Omnichannel

Many scholars have done relevant research on pricing and ordering decision under non-omnichannel, mainly including the decision on pre-sale, perishable or fresh products, returns and other aspects, etc.

Regarding research on pricing and ordering decision under the pre-sale mode, Seref et al. [27] studied the expected effect of consumers comparing pre-orders and deferring to the spot period to make decisions to select the most favorable timing for purchases and how retailers can develop optimal pre-sale prices and purchase quantities based on consumer strategic behavior. Xu et al. [28] constructed a one-order model and a two-order model with initial stock shortages. Sun et al. [29] established a joint pricing and order optimization model for online pre-sale of new products based on robust decision-making behaviors. Chen et al. [30] pointed out that when e-commerce companies adopt a discounted pre-sale strategy, the highest consumer valuation has a positive impact on the decision and expected profit of e-commerce companies. Anily and Hassin [31] consider a deterministic pricing and replenishment model in which the retailer advertises a fixed price and the selling schedule and customers can advance or delay their time of purchase, thus incurring holding or shortage costs.

Regarding pricing and ordering decision of perishable or fresh products, Guan and Li [32], Jia et al. [33], and Maihami et al. [34] studied the joint decision when the demand is random, and the customer is price sensitive. Nie et al. [35] studied the joint decision from the perspective of consumer perception. Rabbani et al. [36] considered the joint decisions when the demand rate depends on price and product quality. Herbon and Khmelnitsky [37] studied the decision when the demand depends on price and time.

The literature on pricing and ordering decision when considering returns includes a study by Su [38] that established an optimal pricing and ordering joint decision model for full and partial refunds based on the customer's uncertainty about the perceived value of the product before purchase. Zhang and Yao [39] introduced customer return to the single-cycle and multi-cycle pricing ordering strategy research of online retailers and pointed out that efforts to control the customer return rate are beneficial to online retailers to obtain high benefits. Akcay et al. [40] studied the retailer's optimal order quantity, optimal pricing, and optimal discount price decision when returns can be resold at a discount and customers can choose between new products and discounted products. Fan and Wang [41] constructed a single-cycle inventory model where returns can be resold under deterministic and random demand for seasonal sales, end-of-sale, and promotional items in online retailing. Noori-daryan and Taleizadeh [42] developed an economic production quantity model in a three-echelon supply chain composed of a supplier, a manufacturer, and a wholesaler under two scenarios.

In addition, Wang et al. [43] studied multi-product pricing and ordering decision, Yi et al. [44], Zhang and Jing [45] considered dual-channel pricing and ordering decision, Tsao and Sheen [46] and Shah et al. [47] studied pricing and ordering decision where demand depends on price and time, Pal et al. [48] considered the decision where price dependents inventory, Sadjadi et al. [49] studied the decision on a two-echelon supply chain, and Fang et al. [50] studied the pricing and ordering decision of green products.

None of the above literature on pricing and ordering decisions for pre-sale, perishables, returns, etc. are all studied the effects of an omnichannel. They did not consider the differences between omnichannel and non-omnichannel and the impact of a newly opened channel on the sustainable development of enterprises and society.

In summary, many authors have studied the operation management of omnichannel BOPS and pricing and ordering decision of non-omnichannel, especially the decision on pricing and ordering under the pre-sale model has important inspiration for this study. However, according to the characteristics of a BOPS channel and the fact that the online return rate is high, the joint decision on pricing and ordering for a BOPS omnichannel has not been studied. In the new retail era, retailers who focus on consumer experience and provide customers with an omnichannel model that is more convenient and with lower shopping costs will create a close, loyal, and strong relationship between them and their customers. There is thus a research gap in this area.

## 3. Model Construction

In this section, we build and solve the expected profit models for retailers based on consumer surplus and purchase ratio using an online channel, store channel, and BOPS channel. We first present the model description and our assumptions in Section 3.1, and then we establish joint optimization models for retailer pricing and ordering before and after opening a BOPS channel in Sections 3.2 and 3.3. We then obtain the optimal pricing and ordering decisions under specific conditions. We also obtain some relevant theories.

### 3.1. Model Description and Assumptions

An omnichannel brand retailer sells a product to consumers through an online channel (simplify O channel), a store channel (simplify S channel), and a BOPS channel. Customers who purchase through the O channel need to bear the cost of shipping and waiting time. Customers who purchase through the BOPS channel and S channel need to bear the inconvenience costs of the store, but when they purchase through the BOPS channel, there is no time cost for finding the product and waiting for checkout and packaging, as there is when shopping through the S channel. Therefore, customers are less inconvenienced when they shop through the BOPS channel.

When customers shop through the S channel, they are able to experience the product before buying, so the return rate is extremely low. On the other hand, when customers shop through the O channel and the BOPS channel, the return rate is high due to lack of experience before purchasing, and customers have the same probability of return under the O channel and the BOPS channel. If a customer is not satisfied with the purchased product from the O channel, he can apply for a return online and courier the product to the retailer at his own expense. The customer needs to pay extra costs such as return shipping and waiting for the refund time [51]. If a customer who purchased through the BOPS channel is not satisfied with the product when he picks it up in store, he can return it directly at the store.

The notations used in the model and their definitions are shown in Table 1.

**Table 1.** Notations and their definitions.

| Notations | Definitions |
|---|---|
| $p_j$ | Sales price before and after opening BOPS (Buy Online and Pickup in Store) channel $(j = 1, 2)$, decision variable |
| $q_j$ | Order quantity before and after opening BOPS channel $(j = 1, 2)$, decision variable |
| $c$ | Cost of the product |
| $o$ | Shopping cost for customers purchase through an O (Online) channel |
| $h$ | Inconvenient cost for customers shopping through an S (Store) channel |
| $l$ | The shopping inconvenience through BOPS channel compared to S channel $(0 < l < 1)$ |
| $\theta$ | The probability of online return $(0 < \theta < 1)$ |
| $v$ | Evaluation of the product, follows uniform distribution in the interval $[v_l, v_h]$ |
| $g(\cdot)$ | Probability density function of consumer psychological valuation $v$ |
| $G(\cdot)$ | Cumulative distribution function of consumer psychological valuation $v$ |
| $X$ | Total demand, follows normal distribution with mean $\mu$ and variance $\sigma^2$ |
| $\alpha_i$ | The proportion of consumers who purchased under channel $i (i = O, S)$ |
| $f(\cdot)$ | Probability density function of total demand $X$ |
| $F(\cdot)$ | Cumulative distribution function of total demand $X$, $\overline{F(\cdot)} \equiv 1 - F(\cdot)$ |
| $\Pi_j$ | The expected profit before and after opening BOPS channel $(j = 1, 2)$ |

To simplify calculations and analysis, we make the following assumptions.

(1)  Since the return rate under the O channel is much higher than the return rate under the S channel [38,52], without loss of generality, we assume that the return rate under S channel is 0.

(2)  Assume that each consumer's psychological valuation of the product is different, and the valuation is a random variable that follows a uniform distribution in the interval $[v_l, v_h]$. Its probability density function and cumulative distribution function are $g(\cdot)$ and $G(\cdot)$ respectively, $\overline{G(\cdot)} \equiv 1 - G(\cdot)$.

The critical psychological valuation of consumers in various channels does not exceed the highest valuation of the distribution interval $v_h$.

(3) Assume that the salvage returned under the O channel and BOPS channel is zero.

(4) The shopping cost of consumers under different channels *o*, *h*, and *lh* is much lower than the lowest estimate of the distribution interval $v_l$.

(5) The shopping cost under an O channel is higher than the inconvenient cost under an S channel, i.e., $o > h$.

(6) The probability of return under an O channel is lower than the convenience of a BOPS channel relative to an S channel, i.e.,$1 - l > \theta$.

According to reference [53], we can get the consumer surplus when purchasing under different channels as shown in Table 2.

**Table 2.** Consumer surplus in three channels.

| Channel | The Probability of Return is not Considered | | The Probability of Returns is Considered |
|---------|------|---------------|-----------------------------------------|
| | **Buy** | **Return or No-buy** | |
| O | $v - p - o$ | $-2o$ | $(1 - \theta)(v - p - o) - 2\theta o$ |
| S | $v - p - h$ | $-h$ | $v - p - h$ |
| BOPS | $v - p - lh$ | $-lh$ | $(1 - \theta)(v - p - lh) - \theta lh$ |

As can be seen from Table 2, when considering online returns $\theta$, the consumer surplus under O channel is $(1 - \theta)(v - p - o) - 2\theta o$, and the condition for consumers to purchase under this channel is $(1 - \theta)(v - p - o) - 2\theta o > 0$. Therefore, the critical psychological value of consumers buying the product under O channel is $v_O = p + o(1 + \theta)/(1 - \theta)$, that is , when $v > v_O$, consumers will buy from O channel. the consumer surplus under S channel is $v - p - h$, and the condition for consumers to purchase under this channel is $v - p - h > 0$. Therefore, the critical psychological value of consumers buying the product under S channel is $v_S = p + h$, that is , when $v > v_S$, consumers will buy from S channel. The consumer surplus under BOPS channel is $(1 - \theta)(v - p - lh) - \theta lh$, and the condition for consumers to purchase under this channel is $(1 - \theta)(v - p - lh) - \theta lh > 0$. Therefore, the critical psychological value of consumers buying the product under BOPS channel is $v_{BOPS} = p + lh/(1 - \theta)$, that is , when $v > v_{BOPS}$, consumers will buy from a BOPS channel.

Next, we will build the retailer's pricing and ordering decision model before and after opening BOPS channel, prove the existence of the optimal solution, and obtain the optimal pricing and ordering decisions, based on the behavior of consumers purchasing from different channels.

### 3.2. Construction and Solution of the Decision Model before Opening a BOPS Channel

Before opening a BOPS channel, the brand retailer only sells their products to consumers through O and S channels. According to the definitions of notations and assumptions in Table 1, the proportion of the consumers who purchase using the O channel is $\alpha_O$, when $v > v_O$, the actual purchase ratio under O channel is $\alpha_O \overline{G}(v_O)$. The proportion of the consumers who purchase under S channel is $\alpha_S$, when $v > v_S$, the actual purchase ratio using the S channel is $\alpha_S \overline{G}(v_S)$. Therefore, the proportion of actual purchasers under the two channels of O and S is as follow.

$$
\begin{aligned}
A_1 &= \alpha_O \overline{G}(v_O) + \alpha_S \overline{G}(v_S) = \alpha_O \frac{v_h - v_O}{v_h - v_l} + \alpha_S \frac{v_h - v_S}{v_h - v_l} \\
&= \frac{\alpha_O[v_h - (p_1 + \frac{1+\theta}{1-\theta}o)] + \alpha_S[v_h - (p_1 + h)]}{v_h - v_l} \\
&= \frac{(\alpha_O + \alpha_S)v_h - \alpha_O \frac{1+\theta}{1-\theta}o - \alpha_S h - (\alpha_O + \alpha_S)p_1}{v_h - v_l}
\end{aligned} \tag{1}
$$

If we assume that $B_1 = (\alpha_O + \alpha_S)v_h - o\alpha_O \frac{1+\theta}{1-\theta} - \alpha_S h$, then $A_1 = \frac{B_1 - (\alpha_O + \alpha_S)p_1}{v_h - v_l}$.

Since the probability density function of the total market demand $X$ is $f(x)$, then the probability density function of the actual demand of the two channels O and S is $\psi_1(x) = \frac{1}{A_1}f(\frac{x}{A_1})$.

Furthermore, we obatin the retailer's expected profit before opening BOPS channel as follows.

$$
\begin{aligned}
\Pi_1(p_1, q_1) &= p_1\int_0^{+\infty}\min\{q_1, x\}\psi_1(x)dx - cq_1 = (p_1 - c)q_1 - p_1\int_0^{\frac{q_1}{A_1}}(q_1 - A_1x)f_1(x)dx \\
&= (p_1 - c)q_1 - p_1\int_0^{\frac{q_1(v_h - v_l)}{[B_1 - (\alpha_O + \alpha_S)p_1]}}[q_1 - \frac{B_1 - (\alpha_O + \alpha_S)p_1}{v_h - v_l}x]f_1(x)dx
\end{aligned}
\tag{2}
$$

According to the above Equation (2), we write $E_1$ as $f(\frac{q_1}{A_1})$, $F_1$ as $\int_0^{q_1/A_1}f(x)dx$, $G_1$ as $\int_0^{q_1/A_1}xf(x)dx$, $M_1$ as $\frac{p_1E_1(\alpha_O + \alpha_S)}{A_1(v_h - v_l)}$, then we can get the following Theorem 1.

**Theorem 1.** *Before opening a BOPS channel, when $M_1(2G_1 + \frac{q_1^2}{A_1^2}M_1) > (1 - F_1 - \frac{q_1}{A_1}M_1)^2$, $\Pi_1(p_1, q_1)$ is a concave function, and the retailer has optimal sales price $p_1^* = \frac{B_1}{2(\alpha_O + \alpha_S)}$ and optimal order quantity $q_1^* = \frac{B_1}{2(v_h - v_l)}F^-(1 - 2c\frac{\alpha_O + \alpha_S}{B_1})$ to maximize expected profit.*

The proof of Theorem 1 appears in Appendix A.

Theorem 1 shows that under certain conditions, the retailer has optimal joint pricing and ordering decisions before opening BOPS channel.

Due to the complexity of the retailer's expected profit model, here we use a two-stage optimization technique to find the optimal solution. First we give $p_1$ to find the optimal solution $q_1^*(p_1)$, then substitute $q_1^*(p_1)$ into the profit function to find the optimal sales price $p_1^*$, and finally substitute $p_1^*$ into $q_1^*(p_1)$ to get the optimal order quantity $q_1^*$.

When $p_1$ is given, we let the first derivative of $\Pi_1(p_1, q_1)$ with respect to $q_1(p_1)$ be 0, that is,

$$
\frac{\partial\Pi_1(p_1, q_1)}{\partial q_1(p_1)} = p_1 - c - p_1\int_0^{\frac{q_1(p)}{A_1}}f(x)dx = 0.
\tag{3}
$$

Then

$$
q_1^*(p_1) = A_1F^-(1 - \frac{c}{p_1}).
\tag{4}
$$

We substitute Equation (4) into expected profit Equation (2), and let the first derivative of $\Pi_1(p_1, q_1^*(p_1))$ with respect to $p_1$ be 0, that is,

$$
\frac{\partial\Pi_1(p_1, q_1^*(p_1))}{\partial p_1} = q_1^*(p_1) - q_1^*(p_1)\int_0^{F^-(1 - c/p_1)}f(x)dx + [A_1 - \frac{p_1(\alpha_O + \alpha_S)}{v_h - v_l}]\int_0^{F^-(1 - c/p_1)}xf(x)dx = 0.
\tag{5}
$$

Thus, we get the optimal sales price as follow.

$$
p_1^* = \frac{B_1}{2(\alpha_O + \alpha_S)}.
\tag{6}
$$

Furthermore, we substitute the optimal price Equation (6) into Equation (4) to get the optimal order quantity as follows.

$$
q_1^* = \frac{B_1 - (\alpha_O + \alpha_S)p_1^*}{v_h - v_l}F^-(1 - \frac{c}{p_1^*}) = \frac{B_1}{2(v_h - v_l)}F^-(1 - 2c\frac{\alpha_O + \alpha_S}{B_1}).
\tag{7}
$$

Therefore, before opening a BOPS channel, there are optimal joint decisions, as shown in Equations (6) and (7), to maximize the retailer's expected profit.

According to the retailer's optimal joint decision before opening a BOPS channel, we can get the following: Lemma 1 and Lemma 2.

**Lemma 1.** *When $p_1$ is given, the optimal order quantity $q_1^*(p_1)$ is positively related to the proportion of purchasers under O channel $\alpha_O$, and the proportion of purchasers under S channel $\alpha_S$ is negatively related to the online return rate $\theta$, the shopping costs under O channel o, the inconvenient costs under S channel h, and the maximum expected valuation of the product $v_h$ is negatively correlated with the minimum expected valuation of the product $v_l$.*

The proof of Lemma 1 appears in Appendix A.

Lemma 1 states that, before opening a BOPS channel, the retailer only sells the product to consumers through O and S channels. When the ratio of purchasers under one channel is determined, the higher the ratio of purchasers under the other channel, the greater the total purchase quantity, and the higher the order quantity. Because the online return rate can reduce the customer's willingness to purchase under the O channel, the shopping costs under the O and S channels also have a certain negative impact on the customer's willingness to purchase. Therefore, the higher the online return rate and the channel shopping costs, the less the order quantity. The higher the consumer's minimum expected value of the product, the more worthwhile the product is, so the larger the sales, the higher the order quantity. Conversely, the higher the consumer's valuation of the product, the higher the price, and the lower the customer's purchase quantity.

**Lemma 2.** *The optimal price $p_1^*$ is negatively related to the proportion of purchasers under online channel $\alpha_O$, the online return rate $\theta$, the shopping costs under O channel o, and the inconvenient costs under S channel h, is positively related to the proportion of purchasers under store channel $\alpha_S$ and the maximum expected valuation of the product $v_h$ and has nothing to do with the minimum expected valuation of the product $v_l$.*

The proof of Lemma 2 appears in Appendix A.

Lemma 2 shows that, due to the impact of the return rate, an appropriate price reduction can achieve the effect of small profits and long sales—that is, the proportion of purchasers under the online channel increases, while the S channel is the opposite. Because the customer decides whether to buy after the experience in the store, when the ratio of purchasers is high, the product itself is worth buying, even if the price increases appropriately. When the online return rate and channel shopping costs are high, a proper price reduction can attract consumers to buy. The higher the consumer's maximum expected valuation of the product, the higher the value of the product itself, and the price should naturally be increased appropriately. Conversely, the product pricing mainly depends on production costs and profit margin and is not affected by the consumer's minimum expected valuation of the product.

### 3.3. Construction and Solution of Decision Model After Opening a BOPS Channel

After opening sBOPS channel, the brand retailer sell the product to consumers through three channel—O, S, and BOPS. According to the definitions of notations and assumptions in Table 1, the proportion of the consumers who purchase under O channel is $\alpha_O$; when $v > v_O$, the actual purchase ratio under the O channel is $\alpha_O \overline{G}(v_O)$. The proportion of the consumers who purchase under S channel is $\alpha_S$, when $v > v_S$, the actual purchase ratio under the S channel is $\alpha_S \overline{G}(v_S)$. The proportion of consumers who purchase under BOPS channel is $1 - \alpha_O - \alpha_S$, when $v > v_{BOPS}$, the actual purchase

ratio under the BOPS channel is $\alpha_S \overline{G}(v_S)$. Therefore, the proportion of actual purchasers under the three channels of O, S, and BOPS as follows.

$$
\begin{aligned}
A_2 \quad &= \alpha_O \overline{G}(v_O) + \alpha_S \overline{G}(v_S) + (1 - \alpha_O - \alpha_S)\overline{G}(v_{BOPS}) \\
&= \alpha_O \frac{v_h - v_O}{v_h - v_l} + \alpha_S \frac{v_h - v_S}{v_h - v_l} + (1 - \alpha_O - \alpha_S)\frac{v_h - v_{BOPS}}{v_h - v_l} \\
&= \frac{\alpha_O[v_h - (p_2 + \frac{1+\theta}{1-\theta}o)] + \alpha_S[v_h - (p_2 + h)] + (1 - \alpha_O - \alpha_S)[v_h - (p_2 + \frac{lh}{1-\theta})]}{v_h - v_l} \\
&= \frac{v_h - p_2 - \alpha_O \frac{1+\theta}{1-\theta}o - \alpha_S h - (1 - \alpha_O - \alpha_S)\frac{lh}{1-\theta}}{v_h - v_l}
\end{aligned}
\tag{8}
$$

We assume $B_2 = \alpha_O o \frac{1+\theta}{1-\theta} + \alpha_S h + (1 - \alpha_O - \alpha_S)\frac{lh}{1-\theta}$, then $A_2 = \frac{v_h - p_2 - B_2}{v_h - v_l}$.

Similar to Section 3.2, since the probability density function of the total market demand $X$ is $f(x)$, then the probability density function of the actual demand of the three channels O, S ,and BOPS is $\psi_2(x) = \frac{1}{A_2}f(\frac{x}{A_2})$.

Furthermore, we obatin the retailer's expected profit after opening a BOPS channel as follows.

$$
\begin{aligned}
\Pi_2(p_2, q_2) \quad &= p_2 \int_0^{+\infty} \min\{q_2, x\}\psi_2(x)dx - cq_2 = (p_2 - c)q_2 - p_2 \int_0^{\frac{q_2}{A_2}} (q_2 - A_2 x)f_2(x)dx \\
&= (p_2 - c)q_2 - p_2 \int_0^{\frac{q_2(v_h - v_l)}{(v_h - p_2 - B_2)}} (q_2 - \frac{v_h - p_2 - B_2}{v_h - v_l}x)f_2(x)dx
\end{aligned}
\tag{9}
$$

According to Equation (9), we write $E_2$ as $f(\frac{q_2}{A_2})$, $F_2$ as $\int_0^{q_2/A_2} f(x)dx$, $G_2$ as $\int_0^{q_2/A_2} xf(x)dx$, $M_2$ as $\frac{p_2 E_2}{A_2(v_h - v_l)}$, we arrive at Theorem 2.

**Theorem 2.** *After opening a BOPS channel, when* $M_2(2G_2 + \frac{q_2^2}{A_2^2}M_2) > (1 - F_2 - \frac{q_2}{A_2}M_2)^2$, $\Pi_2(p_2, q_2)$ *is a concave function, and the retailer has optimal sales price* $p_2^* = \frac{v_h - B_2}{2}$ *and optimal order quantity* $q_2^* = \frac{v_h - B_2}{2(v_h - v_l)}F^-(1 - \frac{2c}{v_h - B_2})$ *to maximize expected profit.*

The proof of Theorem 2 appears in Appendix A.

Theorem 2 shows that, under certain conditions, the retailer has optimal joint pricing and ordering decisions after opening a BOPS channel.

As in Section 3.2, we still use the two-stage optimization technique to find the optimal solution. First, we give $p_2$ to find the optimal solution $q_2^*(p_2)$, then substitute $q_2^*(p_2)$ into the profit function to find the optimal sales price $p_2^*$, and finally substitute $p_2^*$ into $q_2^*(p_2)$ to get the optimal order quantity $q_2^*$.

When $p_2$ is given, we let the first derivative of $\Pi_2(p_2, q_2)$ with respect to $q_2(p_2)$ be 0, that is,

$$
\frac{\partial \Pi_2(p_2, q_2)}{\partial q_2(p_2)} = p_2 - c - p_2 \int_0^{\frac{q_2(p)}{A_2}} f(x)dx = 0.
\tag{10}
$$

Then,

$$
q_2^*(p_2) = A_2 F^-(1 - \frac{c}{p_2}).
\tag{11}
$$

We substitute equation (11) into expected profit equation (9), and let the first derivative of $\Pi_2(p_2, q_2^*(p_2))$ with respect to $p_2$ be 0, that is,

$$
\frac{\partial \Pi_2(p_2, q_2^*(p_2))}{\partial p_2} = q_2^*(p_2) - q_2^*(p_2)\int_0^{F^-(1-c/p_2)} f(x)dx + [A_2 - \frac{p_2}{v_h - v_l}]\int_0^{F^-(1-c/p_2)} xf(x)dx = 0.
\tag{12}
$$

Thus, we get the following optimal sales price:

$$
p_2^* = \frac{v_h - B_2}{2}.
\tag{13}
$$

Furthermore, we substitute the optimal price Equation (13) into Equation (11) to get the optimal order quantity.

$$q_2^* = \frac{v_h - B_2}{2(v_h - v_l)} F^- \left(1 - \frac{c}{p_2^*}\right) = \frac{v_h - B_2}{2(v_h - v_l)} F^- \left(1 - \frac{2c}{v_h - B_2}\right). \tag{14}$$

Therefore, after opening BOPS channel, there are optimal joint decisions as shown in Equations (13) and (14) to maximize the retailer's expected profit.

According to the retailer's optimal joint decision after opening a BOPS channel, we can get Lemma 3 and Lemma 4.

**Lemma 3.** *When $p_2$ is given, the optimal order quantity $q_2^*(p_2)$ is negatively related to the proportion of purchasers using the O channel $\alpha_O$, the online return rate $\theta$, the shopping costs using the O channel o, the inconvenient costs using the S channel h, the inconvenience of customers shopping through the BOPS channel compared to the S channel l, and the maximum expected valuation of the product $v_h$, is positively related to the proportion of purchasers using the S channel $\alpha_S$ and the minimum expected valuation of the product $v_l$.*

The proof of Lemma 3 appears in Appendix A.

Unlike Lemma 1, the optimal order quantity before opening a BOPS channel is positively related to the proportion of purchasers under O channel, the optimal order quantity after opening a BOPS channel is negatively related to the proportion of purchasers using the O channel. This is because after opening a BOPS channel, the sum of the proportion of purchasers using the O, S, and BOPS channels is 1. When the proportion of purchasers using the S channel is determined, as the proportion of purchasers using the O channel increases, the proportion of purchasers using the BOPS channel inevitably decreases, that is, some customers who originally intended to buy on BOPS channel transfer to the S channel to purchase. However, because customers using the O channel need to bear the cost of return shipping and waiting for the refund, which affects their purchase demand, so the total purchase demand decreases with the increase of the proportion of purchasers using the O channel. Correspondingly, the retailer should reduce the order quantity to reduce the losses caused by the returns of the O channel and the BOPS channel. When the proportion of purchasers under the O channel is determined, since the S channel is not affected by the return rate, the higher the proportion of purchasers in this channel, the larger the sales and the order quantity. The correlation between the optimal order quantity and other parameters after opening a BOPS channel is the same as in Lemma 1.

**Lemma 4.** *The optimal price $p_2^*$ is negatively related to the proportion of purchasers under O channel $\alpha_O$, the online return rate $\theta$, the shopping costs under O channel o, the inconvenient costs under S channel h, the inconvenience of customers shopping through BOPS channel compared to S channel l, is positively related to the proportion of purchasers under S channel $\alpha_S$ and the maximum expected valuation of the product $v_h$, and has nothing to do with the minimum expected valuation of the product $v_l$.*

The proof of Lemma 4 appears in Appendix A.

Obviously, Lemma 2 and 4 show that before and after opening a BOPS channel, the correlation between the optimal sales price and other parameters is consistent.

According to Lemma 1 and Lemma 2 in Section 3.2 and Lemma 3 and Lemma 4 in Section 3.3, we can get the following conclusions.

**Conclusion 1.** With the increase in the proportion of O channel purchasers, before opening a BOPS channel, the optimal joint decision of the retailer is to increase the order quantity while lowering the price; after opening a BOPS channel, the retailer's optimal joint decision is to reduce both the price and the order quantity of the product.

**Conclusion 2.** Regardless of whether a BOPS channel is opened or not, as the proportion of S channel purchasers increases, the optimal joint decision of the retailer is to simultaneously increase the price and order quantity of the product; with the increase in online return rate, channel shopping costs,

and the inconvenience of the BOPS channel compared to the S channel, the retailer's optimal joint decision is to simultaneously reduce product price and order quantity; with the increase in the highest valuation of the product, the optimal joint decision of the retailer is to reduce the order quantity while increasing the product price.

Obviously, the decision-making behaviors of the retailer's optimal pricing and ordering in Conclusions 1 and 2 give the direct answer to the second research question. These decisions can enable retailers to meet consumer demand while avoiding a large inventory backlog, thereby maximizing benefits and enhancing the sustainability of healthy business development.

## 4. Sensitivity Analysis

Due to the complexity of the optimal decision operation before and after opening a BOPS channel, in this section, we use the powerful mathematical software Matlab R2016a to design the algorithm and perform numerical calculations under the assumptions in Section 3.1, Theorem 1 in Section 3.2, and Theorem 2 in Section 3.3, and analyze the impact of the main parameters on the optimal decisions, so that the enterprise can improve the enterprise's profit by controlling the changes of each parameter in the management practice. In addition, because the shopping inconvenience costs of the three channels $o$, $h$, and $lh$ are very small compared to the product cost $c$ and the valuations of the product $v_h$ and $v_l$, and the lowest valuation $v_l$ is not related to the optimal pricing, therefore, only the relationship between the optimal decisions and the maximum profit and the four main parameters $\alpha_O$, $\alpha_S$, $\theta$, and $v_h$ are discussed here. According to the real online shopping freight and the customer's transportation cost to the stores and the possible relationship between the product cost and its valuation in the retailer's operating practices, we assume $u = 1000$, $\sigma = 100$, $c = 100$, $o = 8$, $h = 5$, $l = 0.9$, $\alpha_O = 0.2$, $\alpha_S = 0.4$, $\theta = 0.3$, $v_h = 300$, $v_l = 100$, the relevant results are shown in Tables 3–6.

**Table 3.** The impact of $\alpha_O$ on the optimal decisions before and after opening a BOPS channel.

| $\alpha_O$ | $p_1^*$ | $q_1^*$ | $\Pi_1^*$ | $p_2^*$ | $q_2^*$ | $\Pi_2^*$ | $p_2^*-p_1^*$ | $q_2^*-q_1^*$ | $\Pi_2^*-\Pi_1^*$ |
|---|---|---|---|---|---|---|---|---|---|
| 0.1 | 145.86 | 208 | 8859.05 | 147.31 | 698 | 30199.29 | 1.45 | 490 | 21340.25 |
| 0.2 | 145.04 | 276 | 11549.77 | 146.28 | 695 | 29825.90 | 1.24 | 420 | 18276.13 |
| 0.3 | 144.54 | 343 | 14236.17 | 145.66 | 693 | 29454.27 | 1.12 | 350 | 15218.11 |
| 0.4 | 144.21 | 411 | 16920.81 | 145.24 | 691 | 29084.42 | 1.03 | 280 | 12163.61 |
| 0.5 | 143.98 | 478 | 19604.60 | 144.82 | 688 | 28716.34 | 0.84 | 210 | 9111.74 |
| 0.6 | 143.80 | 546 | 22287.89 | 144.40 | 686 | 28350.03 | 0.60 | 140 | 6062.14 |
| 0.7 | 143.67 | 613 | 24970.88 | 143.98 | 683 | 27985.49 | 0.31 | 70 | 3014.62 |

\* Indicates that the value is optimal.

**Table 4.** The impact of $\alpha_S$ on the optimal decisions before and after opening a BOPS channel.

| $\alpha_S$ | $p_1^*$ | $q_1^*$ | $\Pi_1^*$ | $p_2^*$ | $q_2^*$ | $\Pi_2^*$ | $p_2^*-p_1^*$ | $q_2^*-q_1^*$ | $\Pi_2^*-\Pi_1^*$ |
|---|---|---|---|---|---|---|---|---|---|
| 0.1 | 144.21 | 205 | 8506.81 | 146.01 | 695 | 29813.44 | 1.80 | 490 | 21306.63 |
| 0.2 | 145.04 | 276 | 11605.99 | 146.09 | 695 | 29876.63 | 1.05 | 420 | 18270.65 |
| 0.3 | 145.53 | 346 | 14710.85 | 146.16 | 696 | 29939.88 | 0.63 | 350 | 15229.03 |
| 0.4 | 145.86 | 416 | 17818.52 | 146.23 | 696 | 30003.17 | 0.37 | 280 | 12184.65 |
| 0.5 | 146.09 | 487 | 20927.77 | 146.30 | 697 | 30066.52 | 0.21 | 210 | 9138.75 |
| 0.6 | 146.27 | 557 | 24038.00 | 146.37 | 697 | 30129.92 | 0.10 | 140 | 6091.92 |
| 0.7 | 146.40 | 627 | 27148.87 | 146.44 | 697 | 30193.37 | 0.04 | 70 | 3044.50 |

**Table 5.** The impact of $\theta$ on the optimal decisions before and after opening a BOPS channel.

| $\theta$ | $p_1^*$ | $q_1^*$ | $\Pi_1^*$ | $p_2^*$ | $q_2^*$ | $\Pi_2^*$ | $p_2^*-p_1^*$ | $q_2^*-q_1^*$ | $\Pi_2^*-\Pi_1^*$ |
|---|---|---|---|---|---|---|---|---|---|
| 0.1 | 146.70 | 419 | 18231.35 | 147.02 | 701 | 30671.73 | 0.32 | 281 | 12440.39 |
| 0.2 | 146.33 | 418 | 18028.88 | 146.68 | 699 | 30354.17 | 0.34 | 281 | 12325.29 |
| 0.3 | 145.86 | 416 | 17769.10 | 146.23 | 696 | 29946.56 | 0.37 | 280 | 12177.46 |
| 0.4 | 145.22 | 414 | 17423.69 | 145.63 | 693 | 29404.36 | 0.41 | 279 | 11980.67 |
| 0.5 | 144.33 | 411 | 16942.14 | 144.80 | 688 | 28647.97 | 0.47 | 277 | 11705.83 |

**Table 6.** The impact of $v_h$ on the optimal decisions before and after opening a BOPS channel.

| $v_h$ | $p_1^*$ | $q_1^*$ | $\Pi_1^*$ | $p_2^*$ | $q_2^*$ | $\Pi_2^*$ | $p_2^*-p_1^*$ | $q_2^*-q_1^*$ | $\Pi_2^*-\Pi_1^*$ |
|---|---|---|---|---|---|---|---|---|---|
| 300 | 145.86 | 416 | 19007.79 | 146.23 | 696 | 32032.24 | 0.37 | 280 | 13024.45 |
| 350 | 170.86 | 401 | 28174.37 | 171.23 | 670 | 47318.33 | 0.37 | 269 | 19143.96 |
| 400 | 195.86 | 391 | 36917.67 | 196.23 | 653 | 61894.92 | 0.37 | 262 | 24977.25 |
| 450 | 220.86 | 383 | 45383.47 | 221.23 | 640 | 76007.01 | 0.37 | 257 | 30623.54 |
| 500 | 245.86 | 377 | 53653.18 | 246.23 | 630 | 89791.19 | 0.37 | 253 | 36138.01 |

From the data in Table 3, it can be seen that, when the proportion of purchasers using the S channel $\alpha_S$ is determined, the optimal sales price $p_1^*$ and $p_2^*$ decrease with the increase of the proportion of purchasers using the O channel $\alpha_O$ before and after opening a BOPS channel, but the decline is small. The optimal order quantitiy $q_1^*$ increases with the increase of $\alpha_O$ before opening a BOPS channel, and the optimal order quantitiy $q_2^*$ decreases with the increase of $\alpha_O$ after opening a BOPS channel, but the increase or decrease is not significant. This is consistent with the correlation between the optimal decisions and $\alpha_O$ in Lemma 1–4 in Section 3. In addition, the maximum expected profit also increases with the increase of $\alpha_O$ before opening a BOPS channel and gradually decreases with the increase of $\alpha_O$ after opening a BOPS channel.

It can also be seen from Table 3 that, after opening a BOPS channel, the optimal sales price, optimal order quantity, and maximum expected profit increase, but the optimal price does not increase much. In addition, the increase in the optimal sales price first increases and then decreases with the increase of $\alpha_O$, and the increase in the optimal order quantity and the maximum profit decrease with the increase of $\alpha_O$. The increase in the optimal sales price, the optimal order quantity, and the maximum profit all decrease as $\alpha_O$ increases. This shows that the proportion of purchasers using the O channel has little effect on the sales price, but is affected by the online return rate, after the BOPS channel with the same return rate is added, the proportion of purchasers using the O channel has a negative impact on sales and profits.

From the data in Table 4, it can be seen that when the proportion of purchasers using the O channel $\alpha_O$ is determined, the optimal sales price $p_1^*$ and $p_2^*$, the optimal order quantity $q_1^*$ and $q_2^*$, and the maximum expected profit $\Pi_1^*$ and $\Pi_2^*$ increase with the increase of the proportion of purchasers under S channel $\alpha_S$ before and after opening a BOPS channel. This shows that the S channel has the unique experience and service advantages that the O channel lacks, as well as the extremely low return rate that the O channel and the BOPS channel do not have. In addition, the relationship between the optimal price and order quantity decision and $\alpha_S$ is consistent with the correlation between optimal decision and $\alpha_S$ in Lemma 1–4 in Section 3.

It can also be seen from Table 4 that, after opening BOPS channel, the optimal sales price, optimal order quantity and maximum profit all increase, but the optimal price does not increase much. In addition, before and after opening BOPS channel, the increase in the optimal sales price, optimal order quantity, and maximum profit all decrease with the increase in the proportion of purchasers under S channel.

From the data in Table 5, it can be seen that the optimal sales price $p_1^*$ and $p_2^*$, the optimal order quantity $q_1^*$ and $q_2^*$, and the maximum expected profit $\Pi_1^*$ and $\Pi_2^*$ decrease with the increase of the online return rate $\theta$ before and after opening a BOPS channel. This is an obvious fact. Furthermore, due to the characteristics of online ordering using a BOPS channel, it has the same return rate as O channel shopping. Therefore, even when the BOPS channel is opened, as the online return rate

increases, the optimal sales price, order quantity, and maximum profit still decline, but the decline was not significant. In addition, the relationship between the optimal price and order quantity decision and $\theta$ is consistent with the correlation between optimal decision and $\theta$ in Lemma 1–4 in Section 3.

It can also be seen from Table 5 that, after opening a BOPS channel, the optimal sales price, optimal order quantity, and maximum profit increase, but the optimal price does not increase much, and the optimal order quantity and maximum profit increase significantly. That is to say, the opening of a BOPS channel has little impact on sales price, but it has a more obvious impact on order quantity and profits.

From the data in Table 6, it can be seen that the optimal sales price $p_1^*$ and $p_2^*$ and the maximum expected profit $\Pi_1^*$ and $\Pi_2^*$ increase with the increase of the highest valuation of the product $v_h$ before and after opening a BOPS channel, and the increase is significant, but the optimal order quantity $q_1^*$ and $q_2^*$ is the opposite. Among them, the relationship between the optimal price and order quantity decision and $v_h$ is consistent with the correlation between optimal decision and $v_h$ in Lemma 1–4 in Section 3.

It can also be seen from Table 6 that, after opening a BOPS channel, the optimal price, optimal order quantity, and maximum profit can all increase, but the optimal price does not increase much. With the increase of the highest valuation of the product $v_h$, the increase in the optimal price is fixed, the increase in the optimal order quantity gradually decreases, and the increase in the maximum profit gradually increases. This means that the decision of the optimal price mainly depends on the product and the retailer itself. If the product quality is good enough and the brand value is high, the consumer's evaluation of the product also is high. In terms of value, even if the sales decrease as the highest valuation of the product increases, profits still increase.

According to $p_2^* - p_1^* > 0$, $q_2^* - q_1^* > 0$, and $\Pi_2^* - \Pi_1^* > 0$ in Tables 3–6, and the above analysis, we can get the following conclusions.

**Conclusion 3.** The retailer's optimal joint decision and maximum profit can increase after opening a BOPS channel. Among them, the optimal price increases slightly, and the optimal order quantity and maximum profit increases significantly.

**Conclusion 4.** The increase in the optimal price decreases with the increase in the purchase proportion under O and S channels and increases with the increase in the online return rate. The increase in the optimal order quantity and the maximum profit decreases with the increase in the purchase proportion under the O and S channels and the online return rate. The increase in the optimal order quantity decreases with the increase in the maximum psychological valuation of the product, while the maximum profit is the opposite.

**Conclusion 5.** The proportion of purchases under the O and S channels and the online return rate have no significant effect on optimal price, and the maximum psychological valuation of the product has a greater positive impact on optimal price.

Conclusion 3 reflects the impact of opening a BOPS on retailers' optimal decisions and maximum profit. Obviously, when joint pricing and ordering decisions exist, it is profitable for retailers to open a BOPS channel. Conclusions 4 and 5 reflect the impact of various factors on the retailer's optimal decision.

## 5. Conclusions

The omnichannel retailing mode BOPS provides customers with convenient services for online purchases and pickup in store and realizes a seamless purchase experience through online and store channels. This study considers the probability of online returns to calculate the consumer purchase surplus under the three channels of O, S, and BOPS. Based on the proportion of actual purchasers in different channels, the profit models before and after opening a BOPS channel for retailers are constructed to find the optimal price and optimal order quantity to maximize expected profit. The study finds that, under certain conditions, retailers have optimal joint decisions on pricing and ordering before and after opening a BOPS channel, and when the proportion of purchases under the store channel increases, retailers should simultaneously increase product price and order quantity. However, when the online return rate and channel shopping costs increase, retailers should simultaneously

reduce product price and order quantity. Furthermore, the opening of a BOPS channel can help improve the retailer's optimal price and order quantity and maximum profit.

The management enlightenment obtained from the study is that retailers should keep up with industry trends and open up a new BOPS channel in order to meet the diverse needs of consumers for purchasing channels and consumer experience, because a BOPS channel usually bring new demand, increases customer loyalty, and promotes an enterprise's long-term sustainable development. Moreover, no matter how the market share and online return rate of each channel change and whether a BOPS channel is added, the optimal price does not need to be adjusted significantly, but the optimal order quantity can be increased. Furthermore, if retailers focus on the research and development of products and enhance brand value, they do not have to resort to the low-price strategy for discounting and clearing up because the high-value, high-price strategy is more virtuous, which not only helps to save social resources and reduce waste but also plays an important role in the sustainable development of enterprises and society.

Despite the joint decision of optimal price and order quantity presented by this paper, there are still limitations that can be addressed in future research. For example, we assume that the shopping cost using the online channel is higher than the inconvenient cost using the store channel in Section 3.1. However, when a brand retailer sells goods through an online channel, he usually bears the shipping cost when the customer's purchase amount meets a certain amount, so the customer's shopping cost using the online channel is far lower than the inconvenience cost using the store channel. In response to this limitation, future research needs to consider the similarities and differences in the joint optimal decision of omnichannel BOPS pricing and ordering when the online freight is borne by the retailer and the customer, respectively, so as to provide the best sustainable operating strategy for the retailer.

**Author Contributions:** Conceptualization, J.L.; supervision, Q.X.; writing—original draft preparation, J.L.; writing—review and editing, J.L. and Q.X.; project administration, Q.X. and J.L. All authors have read and agreed to the published version of the manuscript.

**Funding:** This research was funded by the Fundamental Research Funds for the Central Universities (Grant No. CUSF-DHU-223201900089; No. CUSF-DHD-2018) and the National Natural Science Foundation of China (Grant No. 71572033).

**Conflicts of Interest:** The authors declare no conflict of interest.

**Appendix A**

**Proof of Theorem 1.** From the brand retailer's expected profit formula (2) before opening a BOPS channel, we know that its second-order partial derivative and Hesse matrix regarding the sales price $p_1$ and order quantity $q_1$ as follows.

$$\frac{\partial^2 \Pi_1(p_1,q_1)}{\partial^2 p_1{}^2} = -\frac{\alpha_O + \alpha_S}{v_h - v_l}\left[2\int_0^{q_1/A_1} xf(x)dx + \frac{p_1 q_1^2(\alpha_O + \alpha_S)}{A_1^3(v_h - v_l)}\right] < 0, \tag{A1}$$

$$\frac{\partial^2 \Pi_1(p_1,q_1)}{\partial^2 q_1{}^2} = -\frac{p_1}{A_1}f\left(\frac{q_1}{A_1}\right) < 0, \tag{A2}$$

$$\frac{\partial^2 \Pi_1(p_1,q_1)}{\partial p_1 \partial q_1} = \frac{\partial^2 \Pi_1(p_1,q_1)}{\partial q_1 \partial p_1} = 1 - \int_0^{q_1/A_1} f(x)dx - \frac{p_1 q_1(\alpha_O + \alpha_S)}{A_1^2(v_h - v_l)}f\left(\frac{q_1}{A_1}\right), \tag{A3}$$

$$|H_1| = \begin{vmatrix} \frac{\partial^2 \Pi_1(p_1,q_1)}{\partial^2 p_1{}^2} & \frac{\partial^2 \Pi_1(p_1,q_1)}{\partial p_1 \partial q_1} \\ \frac{\partial^2 \Pi_1(p_1,q_1)}{\partial q_1 \partial p_1} & \frac{\partial^2 \Pi_1(p_1,q_1)}{\partial^2 q_1{}^2} \end{vmatrix} = M_1\left(2G_1 + \frac{q_1^2}{A_1^2}M_1\right) - \left(1 - F_1 - \frac{q_1}{A_1}M_1\right)^2. \tag{A4}$$

It can be known from equation (A4) that when $M_1(2G_1 + \frac{q_1^2}{A_1^2}M_1) > (1 - F_1 - \frac{q_1}{A_1}M_1)^2$, then $|H_1| > 0$. At this time, $H_1$ is a negative definite matrix, and $\Pi_1(p_1, q_1)$ is a joint concave function of sales price $p_1$ and order quantity $q_1$. The retailer has the optimal decision $p_1^*$ and $q_1^*$ to maximize the retailer's expected profit.

**Proof of Lemma 1.** From Equation (4), we know that the optimal order quantity $q_1^*(p_1)$ is only related to $A_1$. We can directly obtain the correlation between $q_1^*(p_1)$ and other parameters by finding the correlation between $A_1$ and each parameter. Due to $\frac{dA_1}{d\alpha_O} = \frac{(1-\theta)v_h - o(1+\theta)}{2(1-\theta)(v_h - v_l)} > 0, \frac{dA_1}{d\alpha_S} = \frac{v_h - h}{2(v_h - v_l)} > 0, \frac{dA_1}{dh} = -\frac{\alpha_S}{2(v_h - v_l)} < 0, \frac{dA_1}{d\theta} = -\frac{o\alpha_O}{(1-\theta)^2(v_h - v_l)} < 0, \frac{dA_1}{do} = -\frac{(1+\theta)\alpha_O}{2(1-\theta)(v_h - v_l)} < 0, \frac{dA_1}{dv_h} = -\frac{(1-\theta)[(\alpha_O + \alpha_S)v_l - \alpha_S h] - o\alpha_O(1+\theta)}{2(1-\theta)(v_h - v_l)^2} < 0, \frac{dA_1}{dv_l} = \frac{(1-\theta)[(\alpha_O + \alpha_S)v_h - \alpha_S h] - o\alpha_O(1+\theta)}{2(1-\theta)(v_h - v_l)^2} > 0$, thus, the optimal order quantity $q_1^*(p_1) = A_1 F^-(1 - \frac{c}{p_1})$ is positively related to $\alpha_O$, $\alpha_S$, and $v_l$, is negatively related to $\theta$, $o$, $h$, and $v_h$.

**Proof of Lemma 2.** From Equation (6), we know that $\frac{dp_1^*}{d\alpha_O} = -\frac{\alpha_S[o(1+\theta) - (1-\theta)h]}{2(\alpha_O + \alpha_S)^2(1-\theta)} < 0, \frac{dp_1^*}{d\alpha_S} = \frac{\alpha_O[o(1+\theta) - (1-\theta)h]}{2(\alpha_O + \alpha_S)^2(1-\theta)} > 0, \frac{dp_1^*}{d\theta} = -\frac{o\alpha_O}{(\alpha_O + \alpha_S)(1-\theta)^2} < 0, \frac{dp_1^*}{do} = -\frac{\alpha_O(1+\theta)}{2(\alpha_O + \alpha_S)(1-\theta)} < 0, \frac{dp_1^*}{dh} = -\frac{\alpha_S}{2(\alpha_O + \alpha_S)} < 0, \frac{dp_1^*}{dv_h} = \frac{1}{2} > 0, \frac{dp_1^*}{dv_l} = 0$. Therefore, the optimal sales price is negatively related to $\alpha_O$, $\theta$, $o$, and $h$, is positively related to $\alpha_S$ and $v_h$, and has nothing to do with $v_l$.

**Proof of Theorem 2.** From the brand retailer's expected profit Equation (9) after opening a BOPS channel, we can know that its second-order partial derivative and Hesse matrix regarding the sales price $p_2$ and order quantity $q_2$ as follows.

$$\frac{\partial^2 \Pi_2(p_2, q_2)}{\partial^2 p_2^2} = -\frac{1}{v_h - v_l}[2\int_0^{q_2/A_2} xf(x)dx + \frac{p_2 q_2^2}{A_2^3(v_h - v_l)}f(\frac{q_2}{A_2})] < 0 a = 1, \tag{A5}$$

$$\frac{\partial^2 \Pi_2(p_2, q_2)}{\partial^2 q_2^2} = -\frac{p_2}{A_2}f(\frac{q_2}{A_2}) < 0 a = 1, \tag{A6}$$

$$\frac{\partial^2 \Pi_2(p_2, q_2)}{\partial p_2 \partial q_2} = \frac{\partial^2 \Pi_2(p_2, q_2)}{\partial q_2 \partial p_2} = 1 - \int_0^{q_2/A_2} f(x)dx - \frac{p_2 q_2}{A_2^2(v_h - v_l)}f(\frac{q_2}{A_2})a = 1, \tag{A7}$$

$$|H_2| = \begin{vmatrix} \frac{\partial^2 \Pi_2(p_2, q_1)}{\partial^2 p_2} & \frac{\partial^2 \Pi_2(p_2, q_2)}{\partial p_2 \partial q_2} \\ \frac{\partial^2 \Pi_2(p_2, q_2)}{\partial q_2 \partial p_2} & \frac{\partial^2 \Pi_1(p_2, q_2)}{\partial^2 q_2} \end{vmatrix} = M_2(2G_2 + \frac{q_2^2}{A_2^2}M_2) - (1 - F_2 - \frac{q_2}{A_2}M_2)^2 a = 1. \tag{A8}$$

It can be known from Equation (A8) that when $M_2(2G_2 + \frac{q_2^2}{A_2^2}M_2) > (1 - F_2 - \frac{q_2}{A_2}M_2)^2$, then $|H_2| > 0$. At this time, $H_2$ is a negative definite matrix, and $\Pi_2(p_2, q_2)$ is a joint concave function of sales price $p_2$ and order quantity $q_2$. The retailer has the optimal decision $p_2^*$ and $q_2^*$ to maximize the retailer's expected profit.

**Proof of Lemma 3.** From Equation (11), we know that the optimal order quantity $q_2^*(p_2)$ is only related to $A_2$. We can directly obtain the correlation between $q_2^*(p_2)$ and other parameters by finding the correlation between $A_2$ and each parameter. Due to $\frac{dA_2}{d\alpha_O} = -\frac{o(1+\theta) - lh}{2(1-\theta)(v_h - v_l)} < 0, \frac{dA_2}{d\alpha_S} = \frac{h(1-l-\theta)}{2(v_h - v_l)(1-\theta)} > 0, \frac{dA_2}{dv_l} = \frac{(1-\theta)v_h - o\alpha_O(1+\theta) - h[\alpha_S(1-\theta) + l(1-\alpha_O - \alpha_S)]}{2(1-\theta)(v_h - v_l)^2} > 0, \frac{dA_2}{dv_h} = -\frac{(1-\theta)v_l - o\alpha_O(1+\theta) - h[\alpha_S(1-\theta) + l(1-\alpha_O - \alpha_S)]}{2(1-\theta)(v_h - v_l)^2} < 0, \frac{dA_2}{d\theta} = -\frac{2o\alpha_O + lh(1-\alpha_O - \alpha_S)}{2(1-\theta)^2(v_h - v_l)} < 0, \frac{dA_2}{do} = -\frac{(1+\theta)\alpha_O}{2(1-\theta)(v_h - v_l)} < 0, \frac{dA_2}{dh} = -\frac{l(1-\alpha_O - \alpha_S) + \alpha_S(1-\theta)}{2(1-\theta)(v_h - v_l)} < 0, \frac{dA_2}{dl} = -\frac{h(1-\alpha_O - \alpha_S)}{2(1-\theta)(v_h - v_l)} < 0$, thus, the optimal order quantity $q_2^*(p_2) = A_2 F^-(1 - \frac{c}{p_2})$ is negatively related to $\alpha_O$, $\theta$, $o$, $h$, $l$, and $v_h$, is positively related to $\alpha_S$, $v_l$.

**Proof of Lemma 4.** From Equation (13), we know that, $\frac{dp_2^*}{d\alpha_O} = -\frac{o(1+\theta)-hl}{2(1-\theta)} < 0$, $\frac{dp_2^*}{d\alpha_S} = \frac{h(1-l-\theta)}{2(1-\theta)} > 0$, $\frac{dp_2^*}{dh} = -\frac{l(1-\alpha_O-\alpha_S)+\alpha_S(1-\theta)}{2(1-\theta)} < 0$, $\frac{dp_2^*}{d\theta} = -\frac{2o\alpha_O+lh(1-\alpha_O-\alpha_S)}{2(1-\theta)^2} < 0$, $\frac{dp_2^*}{do} = -\frac{\alpha_O(1+\theta)}{2(1-\theta)} < 0$, $\frac{dp_2^*}{dl} = -\frac{h(1-\alpha_O-\alpha_S)}{2(1-\theta)} < 0$, $\frac{dp_2^*}{dv_h} = \frac{1}{2} > 0$, $\frac{dp_2^*}{dv_l} = 0$. Therefore, the optimal sales price $p_2^*$ is negatively related to $\alpha_O$, $\theta$, $o$, $h$, and $l$, is positively related to $\alpha_S$ and $v_h$, and has nothing to do with $v_l$.

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
