# Peer review of "Joint Decision on Pricing and Ordering for Omnichannel BOPS Retailers: Considering Online Returns"

_sustainability, doi:10.3390/su12041539_

Round 1

Reviewer 1 Report

Paper relevance: The omnichannel BOPS provides consumers with a new choice of purchasing. The problem is very interesting: the decision on pricing and ordering for BOPS must be investigated.

The title, abstract, keywords and the Introduction are adequate.

Do the authors have information from a particular company about how much or which share of turnover is made with the BOPS retail mode? Real data is expected. Some market development data?

The authors form three research questions, which are addressed in the paper.

The references (in the literature review) cover the investigated problem. They are appropriate, literature review is correct, but I think that it could be extended with some additional, relevant up-to-date citations to the papers written by non-Chinese authors. Chinese authors cover the majority of all references now (approx. 60 %).

Model description is OK, some minor formatting corrections are needed in Table 1. For example:

( j = 1, 2 ) -> remove unnecessary spaces: (j = 1, 2)

Numerical example should be better introduced and described (not only with the values of some input variables, but also with comments about the selected values) for easier understanding. Lines 363 to 372.

Paper contribution is adequate (appropriate answers are given). Maybe the conclusion could be given in the direct relation to all three research questions.

Author Response

Thank you very much for your comments and suggestions. Our response to reviewer 1 comments, please see the attachment.

Reviewer 2 Report

The article is interesting and good.
There are comments only on the design of the article. For example, on page 4, different line spacing in different paragraphs.

Author Response

Point 1: There are comments only on the design of the article. For example, on page 4, different line spacing in different paragraphs.

Thank you very much for your comments and suggestions. The full text has been adjusted and modified, especially the line spacing in different paragraphs. Please refer to the revised draft for details.

Reviewer 3 Report

Summary: The current manuscript investigates the existence of optimal solutions on pricing and ordering of online retailers, based on the probability of online return, before and after opening Buy Online and Pickup in Store (BOPS). Specifically, this research establishes expected profit models for brand retailers before and after opening BOPS channel based on consumer surplus and purchase ratio under different channels. Overall, this job has important theoretical contribution for the joint optimization decision on pricing and ordering of brand retailers. More broadly, this work fits nicely with emerging work in sustainability testing the most efficient methods for saving costs and time in business research. Therefore, it adheres to the journal’s standards. While interesting, there are remarkable issues that limit its contribution to the literature in sustainability research in its current form. Broadly speaking, remarkable improvements in methodological procedures, bibliography indications and discussions are needed to fully evaluate the validity of the results.

Below, my questions are arranged by the order in which they were motivated while reading the manuscript:  

1. Abstract: The authors are good storytellers. Very clear and direct!

2. Introduction:  While I do agree with the authors regarding the two main streams followed in the literature review, it would be of utmost importance to split them in two sections (i.e. Omnichannel BOPS and joint decisions on pricing and ordering). Also, the authors should make a better effort in contextualizing the research gap with more (recent) literature addressing the main authors’ topic. 

3. Model construction. Interesting explanation on the model construction, descriptions, assumptions and solutions.

4. The authors would need to more deeply explain your contribution to the literature in sustainability research, but also to those studies implemented in the field of business research.

5. Highly simplistic and obvious your limitations, managerial implications and further research.

A Best of luck as you continue with this research.

Author Response

Thank you very much for your comments and suggestions.

Our response to reviewer 3 comments, please see the attachment.

Reviewer 4 Report

The article concerns a significant and relevant topic. The research is very interesting and looks at and describes the model of Pricing and Ordering Omnichannel BOPS Retailers by taking into account the returns of online purchases. The authors have examined the developments of other authors who have done research into pricing.

The article requires improvements in the following areas:

1. The introduction is based on old statistics from 2013, which should be updated.

2. The main contributions of the development presented should not be given before presenting and giving arguments in favor of the model.

3. At the beginning of the model description in point 3, the model should be described in its entirety and all steps should be given as a process. Then a detailed description in points 3.1, 3.2 and 3.3. should follow.

4. The use of the Sensitivity analysis approach is appropriate, but the article would have greater value if the authors justified their decision why this analysis is chosen. It should also be said which one is used because it is known that there is a large number of approaches designed for sensitivity analysis.  In addition, it can be mentioned whether this tool for analyzing has any drawbacks and do they affect the obtained results.

5. Table 1 could be formatted and presented in a better way,

Author Response

Thank you very much for your comments and suggestions. Our response to reviewer 4 comments, please see the attachment.
